# Dietary Phytic Acid, Dephytinization, and Phytase Supplementation Alter Trace Element Bioavailability—A Narrative Review of Human Interventions

**DOI:** 10.3390/nu16234069

**Published:** 2024-11-27

**Authors:** Thiresia Chondrou, Nikoleta Adamidi, Dimosthenis Lygouras, Simon A. Hirota, Odysseas Androutsos, Vaios Svolos

**Affiliations:** 1Laboratory of Clinical Nutrition and Dietetics, Department of Nutrition and Dietetics, School of Physical Education, Sport Science and Dietetics, University of Thessaly, 42100 Trikala, Greece; chondrouthiresia@gmail.com (T.C.); adamidi.nikoleta@gmail.com (N.A.); oandroutsos@uth.gr (O.A.); 2Computer Science Department, Democritus University of Thrace, 65404 Kavala, Greece; dilygou@cs.ihu.gr; 3Larisa Day Care Center of People with Alzheimer’s Disease, Association for Regional Development and Mental Health (EPAPSY), 15124 Marousi, Greece; 4Snyder Institute for Chronic Diseases, Alberta Children’s Hospital Research Institute, Department of Physiology & Pharmacology, Cumming School of Medicine, University of Calgary, Calgary, AB T2N 1N4, Canada; shirota@ucalgary.ca; 5Human Nutrition, School of Medicine Dentistry & Nursing, College of Medical Veterinary and Life Sciences, Glasgow G12 8QQ, UK

**Keywords:** phytase, phytate, phytic acid, iron, zinc, micronutrient, dephytinization, bioavailability, trace element

## Abstract

Background: Phytic acid is abundant in plant-based diets and acts as a micronutrient inhibitor for humans and non-ruminant animals. Phytases are enzymes that break down phytic acid, releasing micronutrients and enhancing their bioavailability, particularly iron and zinc. Deficiencies in iron and zinc are significant public health problems, especially among populations with disease-associated malnutrition or those in developing countries consuming phytic acid-rich diets. This narrative review aimed to summarize findings from human intervention studies on the interactions between phytic acid, phytase, and micronutrient bioavailability. Methods: An extensive PubMed search (1 January 1990 to 8 February 2024) was conducted using MeSH terms (phytic acid, phytase, IP6, “inositol hexaphosphate,” micronutrient, magnesium, calcium, iron, zinc). Eligible studies included human intervention trials investigating the bioavailability of micronutrients following (a) phytase supplementation, (b) consumption of phytic acid-rich foods, or (c) consumption of dephytinized foods. In vitro, animal, cross-sectional, and non-English studies were excluded. Results: 3055 articles were identified. After the title and full-text review, 40 articles were eligible. Another 2 were identified after cross-checking reference lists from included papers, resulting in 42 included articles. Most studies exploring the efficacy of exogenous phytase (9 of 11, 82%) or the efficacy of food dephytinization (11 of 14, 79%) demonstrated augmented iron and zinc bioavailability. Most phytic acid-rich food-feeding studies (13 of 17, 77%) showed compromised iron and zinc bioavailability. Conclusions: Strong evidence supports decreased iron and zinc bioavailability in phytic acid-rich diets and significant improvements with phytase interventions. Studies of longer periods and within larger populations are needed.

## 1. Introduction

Inositol hexaphosphate (IP6), commonly known as phytic acid, is a natural compound predominantly present in cereals, legumes, oilseeds, and nuts, with the richest foods from each category being maize germ, kidney beans, soybean oil, and peanuts, respectively [1]. Its main function is to serve as one of the main forms of phosphorus storage in plant tissues [2]. Despite its importance in plant metabolism, phytic acid is considered an anti-nutrient for both humans and non-ruminant animals [1].

Phytic acid is a potent chelating agent, capable of binding to essential micronutrients such as zinc, iron, calcium, manganese, and magnesium [3]. This results in the formation of insoluble plant salt complexes that significantly reduce the bioavailability of bound minerals, inhibiting their absorption in the gastrointestinal tract [4] and thus leading to potential deficiencies, particularly in populations whose diets are based on foods high in phytic acid [5].

Phytases, also known as myo-inositol hexakisphosphate phosphohydrolase, are enzymes classified under E.C. 3.1.3.8 (the 3-phytase originating from microbe) as well as E.C. 3.1.2.26 (the 6-phytase derived from plants). They catalyze the sequential release of phosphate from phytic acid (IP6), resulting in lower inositol phosphate esters (IP5 to IP1) and free inorganic phosphate (PI) [5]. This process releases orthophosphate groups from the inositol ring of phytic acid, producing intermediate myo-inositol phosphates and free inorganic phosphorus. By breaking down phytic acid, phytase not only makes phosphorus available for bone growth but also releases calcium, magnesium, protein, and lipids, thus enhancing nutrient availability from plant-based diets [6].

Most diets in developing countries mainly consist of plant-based dishes, which have low bioavailability of minerals such as zinc, iron, calcium, and manganese. This is caused by the high deposits of phytic acid contained in these types of diets, which acts as an anti-nutrient and inhibits the absorption of these micronutrients. For example, a cross-sectional study in China found that there was a significant range in phytic acid consumption among residents in six different regions, with intake varying from 648 to 1433 mg per day. Urban residents consumed less phytic acid compared to rural residents, with an intake of 781 mg per day versus 1342 mg per day, respectively [7].

The daily intake of phytic acid varies depending on the country, food culture, food availability, and purchasing habits. It differs significantly between more industrialized nations and low-income regions like South and Southeast Asia, as well as Central Africa. Additionally, there is a distinction between urban areas (large cities) and rural communities in China, where many low-income families live [3]. Furthermore, deficiencies caused by this phenomenon are one of the most important health concerns in many developing countries. For example, children in sub-Saharan Africa who have high needs for iron are often iron deficient, either anemic or not [8]. The local plant-based nutritional habits are phytic acid dense and low on micronutrients, as well as energy-poor foods. As a result of the prominent food insecurity issue, children struggle to meet their energy goals [9]. Also, malnutrition prevention is of high importance, especially in disease-associated malnutrition. For instance, patients with multiple sclerosis are often malnourished and have lost weight unintentionally. Malnutrition has been demonstrated to weaken the immune system, reduce strength, cause fatigue, and impair muscle function, which affects mental function and respiratory muscle strength and increases the risk of infections. Preventing malnutrition is crucial for patients with multiple sclerosis because it can exacerbate existing symptoms like muscle dysfunction, fatigue, and muscle spasms [10]. Another research team investigated the impact of micronutrient deficiencies in people with inflammatory bowel disease (IBD). It seems like there is a notable link between zinc deficiency and the time to subsequent disease relapse in patients with Crohn’s disease (CD). However, it must be further investigated [11]. Acknowledging the significant challenges posed by phytate-induced micronutrient deficiencies to vulnerable populations, one promising approach is the use of exogenous phytase, particularly that derived from *Aspergillus niger* [12].

Given these challenges, there is a pressing need to explore effective strategies to manage phytic acid and enhance micronutrient absorption. This includes evaluating the role of exogenous phytase and other dietary interventions in mitigating the adverse effects of phytic acid. Considering the above-mentioned points, we aimed to review the literature on phytic acid and micronutrient deficiencies, including studies of human trials.

## 2. Materials and Methods

We performed a narrative review searching in the PubMed database (from 1 January 1990 to 8 February 2024), including the terms “Phytic acid” OR phytate OR phytase OR phytin OR IP6 OR “inositol hexaphosphate” OR “inositol hexakisphosphate” or “inositol polyphosphate”) AND (micronutrient OR magnesium OR calcium OR iron OR zinc. The selection process is shown in the PRISMA flow diagram (Figure 1).

A total of 3113 records were identified, and after filtering for the English language, 3055 articles were yielded. Eligible studies included original articles describing interventions that investigated the absorption of micronutrients following (a) the supplementation of exogenous phytase, (b) the consumption of phytic acid, and (c) the consumption of phytase-treated foods. Animal, in vitro, and cross-sectional studies were excluded. We also excluded studies with double or multiple interventions. A total of 2993 studies were excluded, and 2 others were not available in full text.

## 3. Results

After a full-text review of 60 articles, 40 studies fulfilled the eligibility criteria. In addition, 2 studies were identified by hand-searching, resulting in a final inclusion of 42 studies. Three comprehensive evidence tables were made to present the included studies (Appendix A). Phytase units are abbreviated in the current review as FTU, though other abbreviations, including PTU, PU, or U, have also been used in the literature with interchangeable meanings.

The first evidence table presents 11 eligible studies investigating the efficacy of exogenous phytase on micronutrient absorption. Subsequently, the second evidence table presents 17 eligible studies examining the effect of phytic acid presence on micronutrient absorption. Finally, the third evidence table presents 14 studies investigating the consumption of phytase-treated foods on micronutrient absorption. The data from these evidence tables are summarized in Table 1, Table 2, Table 3, which are presented below. 

### 3.1. Phytase Interventions

The studies reviewed took place in various locations that featured different geographical and socio-economic contexts. In Europe, studies were conducted in Switzerland, specifically Zurich and other university settings [13,15,17,20], as well as in Denmark [21] and Sweden [23], providing insights into phytase’s effects in developed countries high-income settings with access to well-controlled research environments. In Africa, research was conducted in Gambia [14], Burkina Faso [16], and Benin [18], which presented data from developing countries with a focus on improving micronutrient absorption among young children through dietary interventions. Additionally, a study in South Africa explored the impact of phytase on school-aged children’s micronutrient status, highlighting regional differences in dietary challenges and intervention strategies [19]. The study conducted in Venezuela provided data on phytase’s impact on iron absorption from a controlled experimental setting, contributing insights from a Latin American context [22]. These diverse locations underscore the broad applicability of phytase in enhancing micronutrient absorption across different populations and dietary environments.

The majority (82%, 9 out of 11) of the reviewed studies consistently reported that phytase supplementation enhanced micronutrient absorption, namely for zinc and iron [14,15,16,17,18,19,20,22,23]. Notably, studies by Bach Kristensen et al., 2005 and Herter-Aeberli et al., 2020, reported no effect of phytase supplementation [13,21], while the remaining nine studies strongly support the hypothesis of micronutrient bioavailability enhancement during phytase supplementation [14,15,16,17,18,19,20,22,23]. For instance, two studies observed significant increases in fractional zinc absorption with phytase supplementation (*p* < 0.001 and *p* < 0.0001), respectively [14,16]. Similarly, another study reported that phytase nearly doubled iron absorption from FeSO4-fortified millet porridge (*p* < 0.001) [18], and Brnić et al. (2014) found that consuming the active phytase enzyme together with a cereal meal increased zinc absorption to the same extent as when the meal was completely dephytinized before consumption [17]. Studies that combined phytase with ascorbic acid showed synergistic effects in improving iron absorption. For example, the authors of one study demonstrated that adding phytase and ascorbic acid to meals significantly increased iron absorption from both FeSO4 and NaFeEDTA (*p* < 0.001) [20], which is reflected in the fact that the combination of phytase and ascorbic acid led to greater iron absorption [18]. Despite the overall positive impact of phytase, there were discrepancies in iron absorption results. Herter-Aeberli et al., 2020, found no significant increase in iron absorption from tef injera with added phytase compared to control meals (*p* > 0.05) [13], contrasting with the significant improvements observed in other studies [15,18]. This inconsistency may arise from differences in food matrices, phytase types, and intervention durations, highlighting the need for context-specific adjustments in phytase application. Bach Kristensen et al. (2005) found that adding phytase to fiber-rich wheat bread did not prevent a decline in iron status, which contrasts with findings from other studies where phytase effectively improved iron absorption [21].

The duration of interventions across the reviewed studies varied significantly, offering a diverse perspective on the effectiveness of phytase on micronutrient absorption. Short-term studies focused on single-day interventions to assess immediate effects on zinc and iron absorption revealed significant short-term increases in nutrient absorption [14,16,18]. In contrast, other studies extended their observations over a few days, offering a slightly broader view of short-term effects [13,15,23]. Longer-duration studies spanned several weeks to months, providing insights into the sustained impact of phytase on iron and zinc status, highlighting the potential for long-term nutritional benefits and the necessity to monitor for any prolonged consumption issues [19,21]. Also, two studies used crossover designs and controlled settings over several weeks to months, offering robust evidence by incorporating repeated measures and washout periods [17,22]. Finally, a study employed a two-day design for each segment to evaluate various combinations of enhancers, emphasizing the flexibility and adaptability of phytase interventions [20].

The type of phytase used across studies was predominantly *Aspergillus niger*, indicating a common preference for this microbial source due to its effectiveness in degrading phytic acid [13,14,15,16,17,18,19,20]. Also, in the study of Sandberg et al., 1996, the authors utilized *Aspergillus niger*, consistent with the previous studies but with a notably high enzyme activity dose. Bach Kristensen et al. (2005) used a phytase produced from *Aspergillus* Oryzae NOVO-L, showing some variation in phytase source but still within the *Aspergillus* genus [21]. Interestingly, Layrisse et al. (2000) diverged by using phytase from wheat, suggesting potential differences in enzymatic activity or stability [22].

The dosage of phytase varied across the studies, reflecting different experimental designs and objectives. Studies by Herter-Aeberli et al., 2020 and Troesch et al., 2011 both used a dose of 380 FTU [13,19], whereas Zyba et al., 2019 used a higher dose of 588 × 2 FTUs [14] and the study by Brnić et al., 2016 used a lower dose of 20.5 FTUs [16]. Notably, three studies had consistent doses, each using 190 FTU [15,17,20]. Cercamondi et al. (2013) used a similar approach with 200 mg/portion, approximately 400 FTU [18], while Bach Kristensen et al. (2005) administered a significantly higher dose of 2500 FTU/100 g [21]. In contrast, Layrisse et al. (2000) used a more moderate dose of 304 U/100 g [22], and Sandberg et al., 1996, applied an exceptionally high activity dose of 4 × 10^7^ FTU/L [23]. These variations highlight the lack of a standardized dosing protocol and suggest that further research is needed to determine optimal dosing for different populations and settings.

The test meals and supplements across the 11 studies demonstrated considerable diversity, reflecting the different research goals and contexts. However, there were notable correlations among studies that used similar or different approaches. Several studies utilized maize-based meals, either as porridge or in combination with other ingredients [13,15,17,20]. These studies consistently investigated the effect of phytase on iron or zinc absorption, demonstrating the relevance of maize as a common dietary staple and the potential benefits of phytase in enhancing nutrient absorption from maize-based foods. Others focused on millet-based porridges, exploring the impact of phytase on zinc and iron absorption in young children [14,16,18]. These studies consistently found that the addition of phytase significantly improved micronutrient absorption, highlighting millet as another key staple where phytase can be beneficial. One study provided a sweetened maize porridge with a micronutrient powder containing iron and zinc [19], while another focused on wheat bread with or without phytase [21]. The differences in these test meals underscore the versatility of phytase across various food forms, although the outcomes varied, with Bach Kristensen et al., 2005, not showing a significant improvement in iron status despite phytase addition [21]. Layrisse et al., 2000, used a basal breakfast with ferrous sulfate or ferrochel [22], and Sandberg et al., 1996, tested white wheat rolls with wheat bran, comparing phytase-active and phytase-deactivated conditions [23].

Finally, the methodological quality and study designs of the 11 studies varied, encompassing randomized controlled trials (RCTs), crossover designs, and controlled experimental studies. Most studies were RCTs [14,16,18,19,21,23], providing robust evidence through randomization and control groups. Other studies employed crossover designs [13,15,17,20], allowing each participant to serve as their own control, thereby reducing inter-individual variability and increasing the reliability of the findings. Layrisse et al. (2000) utilized a controlled, but not randomized, experimental design to test iron absorption under various conditions, adding valuable comparative data [22].

### 3.2. Dietary Phytic Acid Interventions

The 17 included studies encompass a wide range of geographical and socio-economic contexts. Research was conducted in developed countries such as Switzerland [26,30], Sweden [27,35], and the USA [29,33], as well as in low-resource settings such as Guatemala [32,38] and Rwanda [39]. Further, these studies also range in settings, from controlled laboratory environments [34,37] to real-world settings, such as community feeding programs [38] and home-based interventions [32]. Such geographical and socio-economic diversity highlights the widespread applicability of the findings and underscores the global importance of understanding the impact of phytic acid on micronutrient absorption.

Out of the 17 studies, eight studies specifically examined the absorption of iron. More specifically, four of these studies confirmed that phytic acid inhibits iron absorption [26,27,39,40]. For example, high phytic acid content was consistently associated with reduced fractional iron absorption [39,40]. However, other studies, such as Lind et al., 2003 and Hoppe et al., 2018, showed less pronounced effects or mixed results [35,37]. Also, six other studies examined zinc absorption. Half of these studies confirmed that phytic acid inhibits zinc absorption [26,31,32]. For example, diets rich in phytic acid were associated with lower fractional zinc absorption [36]. However, a few studies found no significant difference in zinc absorption with low-phytic acid maize or beans when compared to controls [38,39]. Furthermore, three other studies examined calcium absorption. Most of these studies showed that high-phytic acid content results in reduced calcium absorption [31,33]. For example, calcium absorption was significantly greater from low-phytic acid maize tortillas when compared to control ones [33]. Out of the 17 studies, only 1 study examined the effect of phytic acid on manganese absorption, stating that high phytic acid content inhibits its absorption [26]. Also, a single study assessed the absorption of copper and found no significant impact of phytic acid on copper absorption [30]. These contradictory findings imply that while phytic acid generally affects micronutrient absorption, its effect may be influenced by the dietary context and food matrix.

The duration of the studies’ interventions differed considerably, with intervention periods ranging from one day [26,34] to six months [37]. The studies of Heaney et al. (1991) and Bohn et al. (2004) involved single-day interventions, focusing on the acute effects of phytic acid on micronutrient absorption [26,34]. On the contrary, more longitudinal studies, such as those conducted by Lind et al. (2003) and Hoppe et al. (2018), were conducted over several weeks to months, investigating the effect of phytic acid over greater periods [35,37]. For example, Lind et al. (2003) conducted an intervention period of 6 months [37], while Hoppe et al. (2018) ran interventions for 12 weeks [35]. This diversity reflects different research objectives, from acute to chronic effects, offering a comprehensive insight into the effects of phytic acid on various time scales.

Across all 17 studies, interventions were primarily concerned with controlling the phytic acid content of the test meals. In most cases, this was achieved either by using low-phytic acid bean, maize, or cereal variants [37,38,40] or by adding phytic acid to the meals to test its effects [27,31,34]. A few studies used dehydrated foods to evaluate changes in nutrient absorption [27,30]. These interventions aimed to isolate the effect of phytic acid by comparing low and high phytic acid foods or by adding measured levels of phytic acid to test meals, thus providing evidence of how phytic acid affects micronutrient bioavailability.

There was a wide variety of test meals and diets used in the interventions. Examples include bread made from different types of flour [26], bread prepared with different phytic acid content [27], and maize tortillas with different levels of phytic acid [33]. While some studies used complete meals that included different types of foods, such as beans, maize, and cereals [39,40], other studies focused on specific ingredients, such as iron-fortified infant formulas or cereals with reduced phytic acid [35,37]. This diversity of test meals underlines the adaptability of the research designs to different dietary contexts and provides a broad scope for assessing the effects of phytic acid in different food matrices.

The methodological quality and study design of the 17 studies differed substantially. Almost half of the studies (9 out of 17) used randomized controlled trials (RCTs) [27,31,32,34,35,37,38,39], which are perceived to be robust for assessing causal effects. However, other studies used less stringent designs, such as community-based interventions [38] or non-randomized trials [34]. Designs also differed in terms of blinding and control measures, with some using double blinding [35,37], while others did not specify blinding methods. Such variability in study design both reflects the different technical and organizational challenges faced by the researchers in diverse settings and adds to a well-rounded understanding of the impact of phytic acid on micronutrient absorption.

### 3.3. Dephytinization Interventions

The 14 included studies cover a broad spectrum of geographical and socio-economic settings. Study settings spanned multiple continents, including Europe (France, Switzerland) [43,45], Africa (Malawi, Benin, Rwanda) [46,53,54], Asia (Republic of Korea, Singapore) [49,52], and North America (USA) [42]. These studies involved populations with varying socio-economic backgrounds, from research universities and urban health centers to rural communities with limited access to advanced healthcare. For instance, Manary et al. (2000) and Petry et al. (2014) focused on communities in Malawi and Rwanda, where high levels of dietary phytic acid are prevalent due to staple food sources [53,54]. In contrast, studies like Davidsson et al., 1997 and Zhang et al., 2007 were conducted in more developed settings, such as France and Sweden, with controlled dietary conditions and higher resources for research infrastructure [43,47]. This diversity highlights the global relevance of studying food dephytinization’s impact on micronutrient absorption and the varying challenges faced by different populations.

The studies collectively demonstrate that phytase effectively dephytinizes foods, enhancing micronutrient absorption, albeit the results are mixed. For iron absorption, half of the studies (7 out of 14) consistently show that dephytinization increases iron bioavailability [43,45,46,47,51,54]. Notably, Petry et al. (2014) found that removing up to 95% of phytic acid significantly improved iron absorption from biofortified beans [54], while Hurrell et al. (2003) observed increased iron absorption from cereal porridges with reduced phytic acid content [45]. On the other hand, Davidsson et al., 1997, 1995, reported that, while dephytinization generally improved iron bioavailability, the effect was not always pronounced, and the presence of ascorbic acid further enhanced absorption in some cases [43,44]. For zinc absorption, five studies investigated the impact of phytase treatment [49,50,52,53,54]. More than half of these studies found that phytase treatment improved zinc absorption [52,53,54], with Manary et al. (2000) showing greater fractional absorption in children on a diet with low phytic acid content [53]. However, Petry et al. (2010) and Couzy et al. (1998) reported mixed results, indicating that while phytase treatment enhanced the absorption, the impact varied depending on the presence of polyphenols and the baseline phytic acid levels [49,50]. For manganese absorption, only one study specifically addressed this, showing that dephytinization significantly increased manganese absorption, doubling the fractional absorption compared to non-dephytinized soy formula [44]. This highlights the potential of phytase to enhance the bioavailability of multiple micronutrients, although the effects can vary based on the specific nutrient and context of the diet.

The types, doses, and durations of phytase interventions varied significantly across the studies. Most studies used *Aspergillus niger* phytase, but dosages ranged widely. The doses of phytase used in the studies ranged from as low as 40 FTU/L [51] to as high as 5000 FTU/g [52,53]. For instance, Manary et al. (2000) used 5000 FTU/g of phytase for a short duration of 3–7 days [53], while Davidsson et al. (1995) did not specify the dose but utilized a 1-day intervention [44]. Kim et al. (2007) administered 5000 FTU/g of *Aspergillus niger* phytase over 9 days for high and low phytic acid diets [52]. Some studies, like Hurrell et al., 2003, also did not specify doses but ensured effective phytic acid degradation [45]. The intervention durations overall varied from 1 day [43,44,46,52] to 42 days [53], reflecting different study designs and objectives. This variation underscores the need for standardization in phytase treatment research to better compare outcomes.

The test meals or diets across the studies were diverse and tailored to the specific dietary patterns and research aims of each study. Davidsson et al. (1997) used iron-fortified cereals with native and dephytinized phytic acid content [43], while Petry et al. (2014) employed beans with varying phytic acid levels [54]. Manary et al. (2000) provided corn-plus-soy porridge, with and without phytase, to assess zinc absorption [53]. Zhang et al. (2007) tested oat-based beverages supplemented with different iron compounds and phytase [47]. Other studies included soy formulas [51], fonio porridges [46], and roller-dried cereal porridges [45]. This variety reflects the adaptation of interventions to local dietary habits and nutritional challenges, highlighting the importance of context-specific solutions.

The methodological quality and study designs varied considerably across the 14 studies. Most studies (8 out of 14) employed randomized controlled trials [43,46,47,49,51,52,53,54], which are robust methods for assessing the effects of dietary interventions. For instance, Petry et al. (2014), Kim et al. (2007), and Davidsson et al. (1994) used RCT and crossover designs to evaluate the impact of phytase on iron and zinc absorption [51,52,54]. Other studies used interventional designs with a focus on community-based approaches [48,53]. Despite this, variations in study quality were evident, with some studies lacking detailed information on phytase dosages [43,44,45,46,47,50].

## 4. Discussion

The current narrative review indicates that supplementation with exogenous phytase significantly improves micronutrient absorption, namely for iron and zinc, with most of the reviewed articles confirming this beneficial effect. In contrast, the presence of phytic acid in the diet was associated with reduced absorption of micronutrients, with evidence suggesting that it mainly affects iron and zinc. In addition, dephytinization of foods improves the bioavailability of micronutrients, with most studies showing increased absorption of iron and zinc after removing phytic acid. These findings are especially important in phytic acid-rich diets, where reducing its content can enhance micronutrient absorption and improve overall nutritional status.

Phytic acid consumption varies considerably around the world and is influenced by the dietary habits of each country. In the United Kingdom, phytic acid intake has increased over the decades, from 504–844 mg/day in 1986 [55] to 1436 ± 755 mg/day in 2005 [56]. The average estimated intake in Italy is lower, at 219 mg/day [57]. In India, intake varies widely, with children aged 4–9 years consuming 720–1160 mg/day, while adolescents (10–19 years) consume 1350–1780 mg/day [58]. In Sweden and Finland, intakes for adults following a Western-style diet range from 180 to 370 mg/day [57], while Swedish vegetarians consume significantly higher levels, reaching 1146 mg/day [28]. These data suggest that phytic acid intake varies widely, with the highest intakes being observed in populations with diets rich in plant foods, such as vegetarians and populations in developing countries. Also, the reviewed studies in this paper, while employing different food bases, consistently found that the addition of phytase significantly enhanced iron absorption, reinforcing the effectiveness of phytase in improving iron bioavailability across different meal types.

According to animal studies, microbial phytases appear to improve iron bioavailability by enhancing iron markers, such as plasma transferrin and hemoglobin levels, in rats [59] and pigs [60]. Phytase also significantly increases zinc bioavailability, improving plasma zinc concentration and zinc digestibility in rats [61,62] and pigs [63]. There is also evidence that phytase improves copper absorption, as shown by increased digestibility and urinary excretion of copper in pigs [64]. However, the effect of phytase on manganese absorption is less clear, with a study showing no significant changes in rats [65], while another study reports increased bone manganese concentrations, particularly in pigs [66].

The use of phytase is not only applicable to developing countries but is also of particular importance for the vegetarian and vegan diets that dominate the Western world, as well as for disease-related malnutrition. The presence of phytic acid significantly limits the absorption of minerals such as iron, which can contribute to problems such as iron deficiency anemia. Today, about 25% of the world’s population suffers from anemia, with the largest proportion due to iron deficiency, affecting about 1 billion people [67]. Especially for patients with malnutrition associated with diseases, such as IBD or multiple sclerosis, intake of phytic acid can exacerbate already limited micronutrient absorption [10,11]. Phytase interventions may be an effective strategy to prevent or treat iron deficiency anemia in these populations by improving iron absorption from plant-based foods.

Our review provides insights for addressing micronutrient deficiencies, especially in developing countries where diets are often high in phytic acid and poor in bioavailable micronutrients [68]. The potential of phytase use is emerging as a sustainable and cost-effective strategy to improve the nutritional value of staple foods, which can help improve overall health outcomes in vulnerable communities [69]. In addition, the results of this study can be used in public health programs to inform communities about the effect of phytic acid on nutrient absorption and to promote food preparation methods that reduce phytic acid, such as soaking, fermentation, and sprouting [6]. At the same time, these findings may also be useful in patients with chronic diseases because adequate intake of micronutrients is critical for managing chronic diseases and supporting patients’ quality of life. Iron deficiency, often seen in chronic kidney disease, heart failure, and IBD, can worsen these conditions [70]. At the same time, zinc deficiency, which occurs in malabsorption and liver and kidney disease, negatively affects the immune response and overall recovery [71]. Finally, our study highlights the importance of developing low-phytic acid foods to improve nutrient bioavailability on a global scale.

This narrative review offers significant advantages, extensively covering phytase supplementation, the effect of phytic acid, and food dephytinization on micronutrient absorption. The focus exclusively on intervention studies enhances the validity of the conclusions. Furthermore, the review follows a multifaceted approach, combining three different research directions: investigating the effect of phytase on micronutrient absorption, analyzing the effect of phytic acid as an absorption inhibitor, and evaluating studies using phytase for food dephytinization. This comprehensive approach provides a complete understanding of the effects of each of these factors, helping to identify gaps in the literature and confirming previously known data.

The present narrative review, although extensive, has some weaknesses. First, it was performed using a single search database, and it is not a systematic review, as it does not formulate a clear research question to guide the study selection process. This may limit the generalizability of the results. In addition, the review does not include an assessment of the quality of the studies included, which may reduce the reliability of the overall findings, as it does not consider any methodological weaknesses or limitations of the studies included. While the diversity in study designs enriches the understanding of the dephytinization impact, it also emphasizes the need for consistent reporting and methodological rigor to enhance the comparability and generalizability of findings.

However, despite the widespread acceptance of the safety of phytase, a gap remains regarding the defined optimal dose of phytase that should be used to dephytinize foods with high phytic acid content, such as cereals and legumes. Also, the lack of long-term studies conducted to evaluate the actual benefit of phytase on micronutrient levels in different populations, especially in vulnerable groups such as children and people facing nutritional deficiencies in developing countries, is a shortcoming of this field of research.

Leading organizations, such as the World Health Organization (WHO) and the European Food Safety Authority (EFSA), have addressed the use of phytase in food, guiding safety and dosage. EFSA has approved the safety of 3-phytase produced by *Aspergillus niger* for use in the food industry for dephytinization of cereal products, products of plant origin, and dairy substitutes. The evaluated study showed that there are no safety concerns regarding the genetic modification of the production organism, and the enzyme does not contain any residues from the production organism [72].

EFSA also considered the dietary exposure to the phytase preparation and estimated that exposure levels in children may reach 3.5 mg Total Organic Solids (TOSs) per kilogram body weight per day and in adults 1.2 mg TOS per kilogram body weight, under conservative assumptions WHO [73]. Despite this conservative estimate, the committee found that the safety margin is high (approximately 250), as the no observed adverse effect level from 13-week toxicity studies in rats was 833 mg TOS/kg body weight. Thus, phytase is considered safe in its intended uses, such as in processed foods and food supplements [72].

## 5. Conclusions

This literature review discusses the increased focus on the exogenous phytase as an enhancer of the bioavailability of essential micronutrients, especially iron and zinc, by reducing the phytic acid present in plant foods, which is a common problem. Phytase supplementation has demonstrated high efficacy for nutrient bioavailability, especially in populations of developing countries that predominantly consume diets that are rich in phytic acid and among the vegetarian/vegan populations of the Western world. The present review supports the potential of phytase as a safe option for the prevention of deficiencies in micronutrients—or how to avoid this—while outlining major weaknesses in the current body of data, such as the absence of studies of longer duration and whether there is an optimum dose of phytase. While the results are favorable, it will be necessary to conduct additional studies to verify these outcomes, especially in at-risk populations, such as children and nutritionally deficient individuals in developing countries. There is great potential for phytase use as a low-cost measure for improving public health, but the immediate cornerstone towards its wider use includes guaranteeing the safety of its use, determining the right amounts, and evaluating long-term effects.

## Figures and Tables

**Figure 1 nutrients-16-04069-f001:**
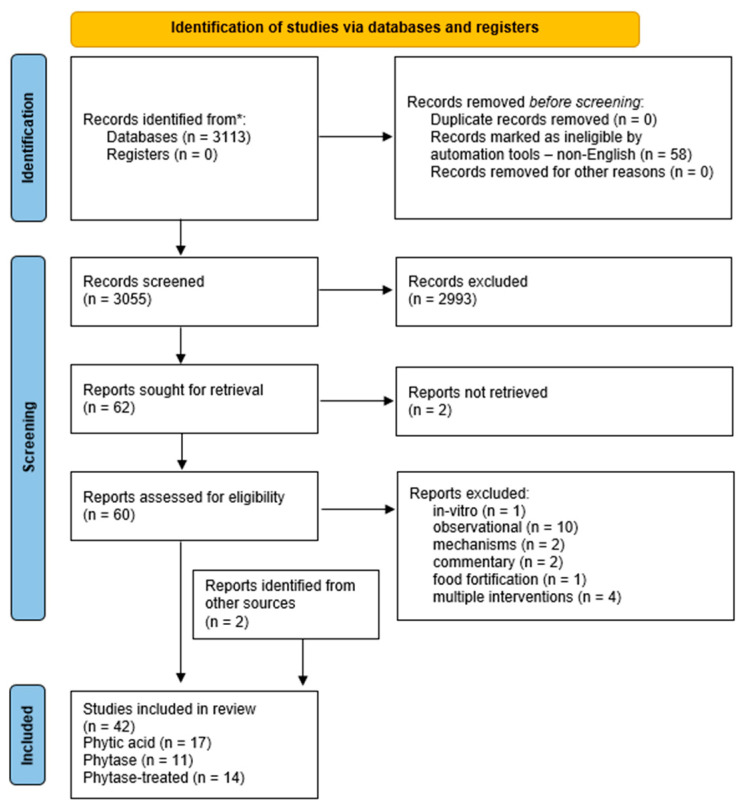
PRISMA 2020 flow diagram for new systematic reviews, which included searches of databases and registers only; * all records were identified through PubMed database.

**Table 1 nutrients-16-04069-t001:** Intervention studies exploring the impact of phytase on micronutrient absorption.

Study	Phytase Type	Daily Dose (FTU)	Duration	Micronutrient	Population	Country	Bioavailability Effect
[13]	*A. niger*	380	3 d	Fe	Ad, *n* = 17	Switzerland	No effect
[14]	*A. niger*	1176	1 d	Zn	Ch, *n* = 26	Gambia	Increase
[15]	*A. niger*	190	2 d	Fe	Ad, *n* = 41	Switzerland	Increase
[16]	*A. niger*	20.5	1 d	Zn	Ch, *n* = 35	Burkina Faso	Increase
[17]	*A. niger*	190	1 d	Zn	Ad, *n* = 60	Switzerland	Increase
[18]	*A. niger*	400	1 d	Fe	Ch, *n* = 18	Natitingou	Increase
[19]	*A. niger*	380	113 d	Fe, Zn	Ch, *n* = 189	South Africa	Increase
[20]	*A. niger*	190	2 d	Fe	Ad, *n* = 101	Switzerland	Increase
[21]	*A. niger*	7500	16 wk	Fe	Ad, *n* = 41	Denmark	No effect
[22]	Wheat	304	1 d	Fe	Ad, *n* = 74	Venezuela	Increase
[23]	*A. niger*	428	2 d	Fe	Ad, *n* = 20	Sweden	Increase

*A. niger*: *Aspergillus niger*, FTU: phytase unit, d: days, wk: weeks, Fe: iron, Zn: zinc, Ch: children, Ad: adults.

**Table 2 nutrients-16-04069-t002:** Intervention studies exploring the impact of phytic acid on micronutrient absorption.

Study	Source of Phytic Acid	Duration	Micronutrient	Population	Country	Bioavailability Effect
[24]	Polenta maize	2 d	Zn	Ad, *n* = 5	USA	Decrease
[25]	High or low phytic acid diet	8 wk	Fe	Ad, *n* = 28	USA	Increase
[26]	Wheat bread, phytic acid-free	2 d	Mn	Ad, *n* = 20	Switzerland	Decrease
[27]	Wheat rolls	4 d	Fe	Ad, *n* = 49	Sweden	Decrease
[28]	Wheat rolls	4 d	Fe	Ad, *n* = 13	Sweden	No effect
[29]	Phytic acid powder	1 d	Fe	Ad, *n* = 30	USA	Decrease
[30]	Dry food	1 d	Cu, Zn	Ad, *n* = 10	Switzerland	Decrease
[31]	White wheat rolls	4 d	Ca, Zn	Ad, *n* = 40	Sweden	Decrease
[32]	High or low PA diet	n/m	Zn	Ad, *n* = 22	Guatemala	Decrease
[33]	Maize	1 d	Ca	Ad, *n* = 5	USA	Decrease
[34]	Soybean	3 d	Ca	Ad, *n* = 16	USA	Decrease
[35]	Wholegrain rye bread	12 wk	Fe	Ad, *n* = 55	Sweden	Decrease
[36]	Different 1-day menus	8 d	Zn	Ad, *n* = 10	USA	Decrease
[37]	Milk-based cereal and porridge	6 mo	Fe, Zn	Ch, *n* = 267	Sweden	No effect
[38]	Maize	10 wk	Zn	Ch, *n* = 60	Guatemala	No effect
[39]	Beans	3 wk	Fe	Ad, *n* = 25	Rwanda	Decrease
[40]	Bean porridge	2 d	Fe	Ad, *n* = 20	Switzerland	Decrease

n/m: not mentioned, d: days, mo: months, wk: weeks, Fe: iron, Zn: zinc, Mn: manganese, Ca: calcium, Cu: copper, Ch: children, Ad: adults.

**Table 3 nutrients-16-04069-t003:** Intervention studies exploring the impact of food dephytinization on micronutrient absorption.

Study	Phytase Type	Duration	Micronutrient	Population	Country	Bioavailability Effect
[41]	n/m	1 d	Fe	Ad, *n* = 18	UK	Increase
[42]	*A. niger*	1 d	Zn, Fe	Ch, *n* = 9	USA	Increase
[43]	*A. niger*	2 d	Fe	Ch, *n* = 12	France	No effect
[44]	*A. niger*	1 d	Mn	Ad, *n* = 16	Switzerland	Increase
[45]	*A. niger*	2 d	Fe	Ad, *n* = 78	Switzerland	Increase
[46]	Wheat	1 d	Fe	Ad, *n* = 42	Benin	Increase
[47]	*A. niger*	2 d	Fe	Ad, *n* = 15	Sweden	Increase
[48]	n/m	40 d	Zn, Fe	Ch, *n* = 10	Malawi	Increase
[49]	*A. niger*	2 d	Fe	Ad, *n* = 97	Singapore	Increase
[50]	n/m	2 d	Zn	Ad, *n* = 39	Switzerland	Increase
[51]	Finase S40	2 d	Fe	Ch, *n* = 10	France	Increase
[52]	*A. niger*	9 d	Zn	Ad, *n* = 17	Republic of Korea	No effect
[53]	*A. niger*	3–7 d	Zn	Ch, *n* = 23	Malawi	Increase
[54]	*A. niger*	42 d	Fe	Ad, *n* = 22	Rwanda	Increase

n/m: not mentioned, *A. niger*: *Aspergillus niger*, d: days, Fe: iron, Zn: zinc, Ch: children, Ad: adults.

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
