# Peer review of "Dietary Phytic Acid, Dephytinization, and Phytase Supplementation Alter Trace Element Bioavailability—A Narrative Review of Human Interventions"

_nutrients, 2024, doi:10.3390/nu16234069_

Round 1
Reviewer 1 Report
Comments and Suggestions for Authors
The manuscript present a focused review on an important subject of current interest, well written, which can be read with pleasure.
Although it is not a fully systematic review, PRISMA guidelines for the selection of sources were applied and the flowchart is reported as a Supplementary Table 2.
Limitations of the study are listed. Conclusions are scientifically sound.
Limiting the search to Pubmed is a somewhat weak point of the study (though guaranteeing a better reliability of data). Many journals publishing relevant studies in food science are not covered by Pubmed.
Lines 118-124: Perhaps this paragraph should be moved to the “Results” section.
Tables: According to my knowledge, “wk” rather than “w” is a standard abbreviation for week
Tables 1 and 2: Please consider: perhaps geographic information about population studied can be added? Scrutinously prepared Table S1 provides detailed data but on the country where the study was performed rather than on the population studied (or am I wrong?)
Line 409: “Our review provides valuable insights”, a dose of modesty would be appropriate. Let readers evaluate the value of the review
Author Response
Reviewer 1
Comments and Suggestions for Authors:
Comments: The manuscript presents a focused review on an important subject of current interest, well written, which can be read with pleasure. Although it is not a fully systematic review, PRISMA guidelines for the selection of sources were applied and the flowchart is reported as a Supplementary Table 2. Limitations of the study are listed. Conclusions are scientifically sound. Limiting the search to Pubmed is a somewhat weak point of the study (though guaranteeing a better reliability of data). Many journals publishing relevant studies in food science are not covered by Pubmed.
Response: We would like to thank the reviewer for the positive feedback. Since this is a narrative review, the focus was on a curated synthesis rather than an exhaustive search. We selected PubMed due to its reliability and relevance for the study's scope. However, we acknowledge the limitation and will consider incorporating additional databases in future studies to enhance coverage. This study limitation is now added within discussion to ensure transparency.
Comments: Lines 118-124: Perhaps this paragraph should be moved to the “Results” section.
Response: Thanks for your comment. This section of the manuscript is now moved to the Results section.
Comments: Tables: According to my knowledge, “wk” rather than “w” is a standard abbreviation for week
Response: This is now changed throughout the manuscript.
Comments: Tables 1 and 2: Please consider: perhaps geographic information about population studied can be added? Scrutinously prepared Table S1 provides detailed data but on the country where the study was performed rather than on the population studied (or am I wrong?)
Response: Thank you for your comment. We agree and have now added a column to Tables 1, 2 and 3 providing the country of the study conduct. Also note that Supplementary Tables include a broad description of the population in each study (a. number of participants; b. heatlhy or disease population; c. age group i.e. inflants, adults or children).
Comments: Line 409: “Our review provides valuable insights”, a dose of modesty would be appropriate. Let readers evaluate the value of the review
Response: Thanks for your comment. We have now removed the word “valuable” to downgrade this sentence.
Reviewer 2 Report
Comments and Suggestions for Authors
I congratulate the authors on their interesting research. My main question concerns the option to perform the literature search only in PubMed.
The type of review needs to be mentioned in the title, the abstract, and the whole manuscript.
The abstract is too extensive once the word limit according to the journal’s guidelines is 250.
Considering the nature of your research, you should include some illustrations. It always benefits a review paper and attracts potential readers.
Why did you only consider one database to conduct your search? In my point of view, you should also consider other relevant and international databases such as Web of Science or Google Scholar.
The PRISMA flowchart should be included as a figure, not as supplementary material.
Some excerpts of the Results section should be moved to Discussion. Please, revise it.
Author Response
Reviewer 2
Comments and Suggestions for Authors:
Comments: I congratulate the authors on their interesting research. My main question concerns the option to perform the literature search only in PubMed. Why did you only consider one database to conduct your search? In my point of view, you should also consider other relevant and international databases such as Web of Science or Google Scholar.
Response: Similar to Reviewer 1, we would like to thank Reviewer 2 for the positive feedback. Since this is a narrative review, the focus was on a curated synthesis rather than an exhaustive search. We selected PubMed due to its reliability and relevance for the study's scope. However, we acknowledge the limitation and will consider incorporating additional databases in future studies to enhance coverage. This study limitation is now added within discussion to ensure transparency.
Comments: The type of review needs to be mentioned in the title, the abstract, and the whole manuscript.
Response: Thanks for your comment. We agree and we have now added the term Narrative Review throughout the manuscript. Our paper synthesizes primary studies and explores this through description rather than statistics, hence we feel Narrative Review is the best type to refer to.
Comments: The abstract is too extensive once the word limit according to the journal’s guidelines is 250.
Response: Thanks for your comment, we have now reduced the length to satisfy the limit of 250 words.
Comments: Considering the nature of your research, you should include some illustrations. It always benefits a review paper and attracts potential readers.
Response: Thank you for your valuable suggestion, which we believe gives us the opportunity to improve our manuscript. We have now included a graphical abstract in the manuscript to enhance visual appeal and effectively communicate our findings.
Comments: The PRISMA flowchart should be included as a figure, not as supplementary material.
Response: Thank you for pointing this out. We agree and have now moved the PRISMA flowchart from the supplementary material to the main manuscript as a figure, as requested.
Comments: Some excerpts of the Results section should be moved to Discussion. Please, revise it.
Response:Thank you for your helpful feedback. Based on your suggestion, we have moved some of the excerpts from the Results section to the Discussion to improve the clarity and flow of the paper.